# Synthesis and Preliminary Characterization of Putative Anle138b-Centered PROTACs against α-Synuclein Aggregation

**DOI:** 10.3390/pharmaceutics15051467

**Published:** 2023-05-11

**Authors:** Martina Pedrini, Angelo Iannielli, Lorenzo Meneghelli, Daniele Passarella, Vania Broccoli, Pierfausto Seneci

**Affiliations:** 1Chemistry Department, Milan University, Via Golgi 19, 20133 Milan, Italy; martina.pedrini@unimi.it (M.P.);; 2Neuroscience Institute, Consiglio Nazionale delle Ricerche (CNR), Via Olgettina 58, 20132 Milan, Italy; 3Stem Cells and Neurogenesis Unit, San Raffaele Hospital, Via Olgettina 58, 20132 Milan, Italy

**Keywords:** Anle138b, α-Synuclein, Parkinson’s disease, neurodegeneration, PROTACs, oncology, medicinal chemistry, protein degradation, ubiquitin proteasome system, Thioflavin T, *4xSNCA* DANs

## Abstract

The search for disease-modifying agents targeted against Parkinson’s disease led us to rationally design a small array of six Anle138b-centered PROTACs, **7a**,**b**, **8a**,**b** and **9a**,**b**, targeting αSynuclein (αSyn) aggregates for binding, polyubiquitination by the E3 ligase Cereblon (CRBN), and proteasomal degradation. Lenalidomide and thalidomide were used as CRBN ligands and coupled with amino- and azido Anle138b derivatives through flexible linkers and coupling reactions (amidation, ‘click’ chemistry). Four Anle138b-PROTACs, **8a**,**b** and **9a**,**b,** were characterized against in vitro αSyn aggregation, monitoring them in a Thioflavin T (ThT) fluorescence assay and in dopaminergic neurons derived from a set of isogenic pluripotent stem cell (iPSC) lines with SNCA multiplications. Native and seeded αSyn aggregation was determined with a new biosensor, and a partial correlation between αSyn aggregation, cellular dysfunctions, and neuronal survival was obtained. Anle138b-PROTAC **8a** was characterized as the most promising αSyn aggregation inhibitor/degradation inducer, with potential usefulness against synucleinopathies and cancer.

## 1. Introduction

A constant increase in life expectancy in western countries is associated, inevitably, with the onset of aging-related neurodegenerative diseases (NDs), such as Alzheimer’s (AD) and Parkinson’s disease (PD). None of the approved treatments heal these debilitating and untreatable pathologies [1]; novel small molecule disease modifiers with innovative mechanisms of action are a popular and pursued research aim [2].

The aggregation of neuronal, misfolded proteins into the formation of insoluble fibrillar polymeric structures is a feature shared by most NDs. These aggregates accumulate in the cytosolic and/or nuclear space of neurons (i.e., tau tangles [3]) or in the extracellular CNS space (i.e., amyloid fibrils [4]). Their spreading in ND-specific CNS areas affects neuronal functionality and, eventually, may cause neuronal cell death [5].

α-Synuclein (αSyn) [6] is a small, 140 amino acid intrinsically disordered protein (IDP), which forms stable secondary structures upon interaction with lipid membranes [7]. It is found in intra-cytoplasmic protein inclusions from brain tissues of patients suffering of synucleinopathies, including familial and sporadic PD, Lewy body dementia, and multi-system atrophy [8]. αSyn becomes prone to aggregation either because of congenital mutations (familial PD) [9] or after post-translational modifications such as phosphorylation, oxidation, and proteolytic cleavage in sporadic PD and other synucleinopathies [10]. Remarkably, αSyn aggregation and upregulation have been recently connected to tumor growth and aggressiveness for melanoma [11] and meningioma [12]. Thus, PROTAC-driven αSyn degradation may be relevant also in oncology.

Therefore, small molecule αSyn binders able to inhibit its oligomerization and aggregation and, therefore, reduce its toxicity, have become a focus for the scientific community [13].

Among known αSyn small molecule binders, diphenylpyrazole Anle138b [14,15] resulted from an HTS as an inhibitor of misfolded protein aggregation, effective in inhibiting the aggregation of both prions and αSyn in vitro and in vivo [16,17,18].

A recently proposed approach to fight NDs entails the elimination of misfolded protein oligomers before they aggregate, thus preventing their fibrillization, by taking advantage of the ubiquitin–proteasome system (UPS) [19,20]. The UPS is a major cellular degradation mechanism, to dispose of soluble misfolded proteins. It involves “labeling” of nuclear/cytosolic misfolded proteins with ubiquitin (UBQ), followed by the recognition and degradation of ubiquitinated proteins of interest (POIs) by the multi-subunit proteolytic proteasome complex [21].

To hijack the UPS against neuronal misfolded proteins, which would not be UPS substrates, one must assemble a POI-UBQ-targeted hybrid called Proteolysis-Targeting Chimera (PROTAC) [22,23]. This heterobifunctional molecule consists of a POI-selective ligand and of an UBQ E3 ligase ligand, targeting an enzyme class capable of ubiquitinating proteins and directing them to proteasomal degradation. The ligands must be connected by an adjustable linker to ensure preservation of both binding properties in a ternary POI/PROTAC/E3 ligase complex. If so, the proximity of the POI to the UBQ E3 ligase causes POI polyubiquitination, release from the ternary complex and degradation by the proteasome [20]. PROTACs centered around peptide [24] or small molecule αSyn ligands [25], leading to UPS-driven αSyn degradation, have been reported recently.

In this work, we selected Anle138b as a potent in vivo active αSyn ligand currently undergoing clinical trials and thalidomide/lenalidomide scaffolds as CRBN E3 ligase ligands [26]. Our synthetic strategy, including the choice of multiple linker connections either through an amide bond (amidation) or a triazole bridge (‘click chemistry’), was designed to synthesize a small array of prospective PROTACs to hijack and dispose of undruggable and overexpressed αSyn against neurodegeneration or cancer. An in vitro αSyn aggregation assay was set up to test our putative PROTACs as Anle138b-centered binders of αSyn monomers/oligomers. Then, dopaminergic neurons (DANs) derived by a set of isogenic iPSC lines with *SNCA* multiplications were chosen to monitor native and seeded αSyn aggregation in an aggressive model of early parkinsonism [27] for in vitro active PROTACs. Our results, including the selection of the promising, neuronally active putative PROTAC **8a** are presented here.

## 2. Materials and Methods

### 2.1. Medicinal Chemistry

#### 2.1.1. General

Oven-dried glassware was used to carry out chemical reactions, and dry solvents under a nitrogen atmosphere were employed. Solvents were purchased from Sigma Aldrich and used as such. Chemical reagents were purchased from Sigma Aldrich, Fluorochem and TCI and checked for integrity before using them. Purification of intermediates and final products was carried out by flash chromatography using high purity grade silica gel (Merck Grade, pore size 60 Å, 230–400 mesh particle size, Sigma-Aldrich, Milan, Italy) as a stationary phase. Alternatively, purification was performed by a BIOTAGE^®^ system using Biotage KP-SIL cartridges (4, 10, or 25 g) for direct phase chromatography or Biotage KP-C_18_-HS cartridges (6, 12, or 30 g) for reverse phase chromatography. Reaction monitoring by thin layer chromatography (TLC) entailed Merck-precoated 60F_254_ plates. Reactions were monitored by TLC on silica gel, using UV light at 254 nm as a direct detection method, or by charring either with a phosphomolybdic acid ethanolic solution, with a potassium permanganate solution or with a ninhydrin solution. ^1^H-NMR and ^13^C-NMR spectra were recorded in either acetone-d6, CDCl_3_, CD_3_OD, DMSO-d6, or pyridine-d5, depending on compounds’ solubility, on Bruker DRX-400 and Bruker DRX-300 instruments. Chemical shifts (δ) for proton and carbon signals are quoted relatively to tetramethylsilane as an internal standard and expressed in parts per million (ppm). Ultra-high performance liquid chromatography/mass spectrometry analysis (UPLC/MS) was performed using an Acquity UPLC/MS System equipped with a tunable ultraviolet (TUV) detector, a single quadrupole (SQD) mass spectrometer and ACQUITY UPLC BEH SHIELD RP18 columns (2.1 × 100 mm, id = 1.7 μm).

The synthesis of Anle138b m-Triazole Connected Lenalidomide PROTAC **8a** and of the intermediates leading to its synthesis is reported in Materials and Methods here. The synthesis of PROTACs **7a**,**b**, **8b** and **9a**,**b**, and of the intermediates leading to their synthesis is reported in the Appendix A. ^1^H- and ^13^C-NMR spectra of all synthesized intermediates and PROTACs are provided in the Appendix A.

#### 2.1.2. Synthesis of m-Nitro-Substituted Diphenylpyrazole **3a**

3′-Nitroacetophenone (1.00 g, 6.06 mmol, 2.0 eq.) was dissolved in dry toluene (4 mL) under N_2_ atmosphere and cooled to 0 °C under stirring. A 1M LiHMDS solution in THF (6.4 mL, 6.40 mmol, 2.1 eq.) was added quickly via a syringe, and the resulting solution was stirred for 5 min. Then, acyl chloride (560 mg, 3.03 mmol, 1.0 eq.) dissolved in dry toluene (3.6 mL) was added dropwise to the stirred enolate solution at 0 °C. The reaction mixture was warmed to r.t and stirred for another 10 min, monitoring via TLC (eluent mixture: 6:4 *n*-hex/EtOAc, developed in phosphomolybdic acid ethanolic solution). After reaction completion, AcOH (6.1 mL, 106 mmol, 35 eq.) was added in one portion, followed by EtOH (15.2 mL), THF (7.6 mL), and a 65% aqueous hydrazine solution (7.7 mL, 103 mmol, 34 eq.). The resulting solution was heated at reflux for 1 h, monitoring via TLC (eluent mixture: 6:4 *n*-hex/EtOAc, developed in phosphomolybdic acid ethanolic solution). After reaction completion and quenching with 1M NaOH (20 mL), the collected organic phases were extracted with EtOAc (70 mL), dried over Na_2_SO_4_, and concentrated in vacuo. The crude solid was purified via flash chromatography (eluent mixture: 7:3 *n*-hex/EtOAc), affording pure compound 3a (2.18 mmol, 72% yield).

##### Analytical Characterization


**3a:**




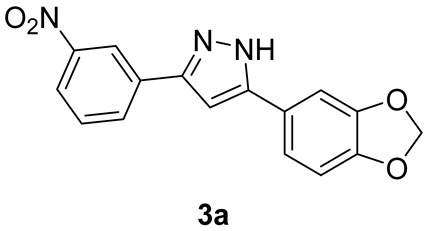



**^1^H NMR** (400 MHz, acetone-d6): δ (ppm) = 12.68 (bs, 1H), 8.71 (t, *J* = 2.0 Hz, 1H), 8.31 (ddd, *J* = 7.8, 1.7, 1.0 Hz, 1H), 8.19 (ddd, *J* = 8.2, 2.3, 1.0 Hz, 1H), 7.74 (t, *J* = 8.0 Hz, 1H), 7.43–7.36 (m, 2H), 7.26 (s, 1H), 6.96 (d, *J* = 8.8 Hz, 1H), 6.07 (s, 2H).

**^13^C NMR** (101 MHz, acetone-d6): δ (ppm) = 132.2, 131.1, 123.0, 120.6, 120.3, 109.6, 106.8, 102.5, 100.9. Coherent with published data [28].

**MS (ESI^+^)**, m/z calcd for C_16_H_11_N_3_O_4_: 309.07, found 310.03 [M + H^+^].

#### 2.1.3. Synthesis of Anle138b-Based m-Anilino Derivative **1a**

*m*-Nitro-derivative **3a** (600 mg, 1.94 mmol, 1.0 eq.) was suspended in 5:1 EtOH/H_2_O (19.4 mL) under stirring at r.t. Fe powder (2.20 g, 38.8 mmol, 20.0 eq.) and a 6M H_2_SO_4_ aqueous solution (1.6 mL, 9.7 mmol, 5.0 eq.) were sequentially added to the reaction mixture, which was then heated and refluxed under stirring for 1 h. The reaction was monitored via TLC (eluent mixture: 4:6 *n*-hex/EtOAc, developed in phosphomolybdic acid ethanolic solution) and, after its completion, the hot solution was filtered over Celite pad and washed with hot EtOH (50 mL). Saturated aqueous NaHCO_3_ (20 mL) was then added to quench the acid. The collected organic phases were extracted with EtOAc (50 mL), dried over Na_2_SO_4_, and concentrated in vacuo. The crude yellowish/orangish solid was purified via flash chromatography (eluent mixture: 4:6 *n*-hex/EtOAc), affording pure target **1a** (1.27 mmol, 65% yield) as a pinkish solid.

##### Analytical Characterization


**1a:**




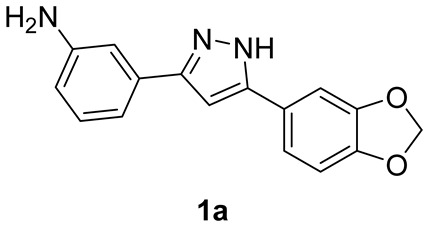



**^1^H NMR** (400 MHz, acetone-d6): δ (ppm) = 12.27 (s, 1H), 7.41–7.33 (m, 2H), 7.14 (t, *J* = 1.5 Hz, 1H), 7.12 (t, *J* = 7.7 Hz, 1H), 7.07 (dt, *J* = 7.6, 1.4 Hz, 1H), 6.90 (d, *J* = 8.6 Hz, 1H), 6.88 (s, 1H), 6.66 (ddd, *J* = 7.8, 2.3, 1.2 Hz, 1H), 6.03 (s, 2H), 4.70 (s, 2H).

**^13^C NMR** (101 MHz, acetone-d6): δ (ppm) = 149.7, 149.0, 148.2, 130.3, 119.9, 115.0, 114.9, 112.1, 109.2, 106.6, 102.1, 99.8. Coherent with published data [28].

**MS (ESI^+^)**, m/z calcd for C_16_H_13_N_3_O_2_: 279.10, found 280.25 [M + H^+^].

#### 2.1.4. Synthesis of Anle138b-Based m-Azido **2a**

*m*-Anilino derivative **1a** (176 mg, 0.63 mmol, 1.0 eq.) was suspended in 1:1 HCl/AcOH (1.3 mL) and cooled to 0 °C under stirring. A 1.8M NaNO_2_ aqueous solution (1 mL, 1.89 mmol, 3.0 eq.) was added in one portion, and the reaction mixture was stirred at 0 °C for 15 min. Then, a 1.8M NaN_3_ aqueous solution (1 mL, 1.89 mmol, 3.0 eq.) was slowly added under vigorous stirring. The reaction solution was warmed to r.t. and stirred for an additional 1 h. The reaction was monitored via TLC (eluent mixture: 6:4 *n*-hex/EtOAc, developed in phosphomolybdic acid ethanolic solution) and, after its completion, was diluted with saturated aqueous NaHCO_3_ (3 mL). The aqueous phase was extracted with EtOAc (4 × 10 mL), and the collected organic phases were dried over Na_2_SO_4_, and concentrated in vacuo. The crude orange solid was purified via flash chromatography (eluent mixture: 6:4 *n*-hex/AcOEt), affording pure *m*-azido target **2a** (0.54 mmol, 86% yield) as a brownish solid.

##### Analytical Characterization


**2a:**




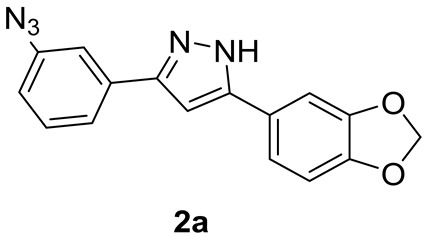



**^1^H NMR** (400 MHz, acetone-d6): δ (ppm) = 7.68 (d, *J* = 7.8 Hz, 1H), 7.58 (t, *J* = 1.9 Hz, 1H), 7.46 (t, *J* = 7.9 Hz, 1H), 7.40–7.33 (m, 2H), 7.08 (s, 1H), 7.03 (dd, *J* = 7.9, 2.4 Hz, 1H), 6.91 (d, *J* = 8.5 Hz, 1H), 6.04 (s, 2H).

**^13^C NMR** (101 MHz, acetone-d6): δ (ppm) = 149.1, 148.7, 148.5, 148.2, 141.4, 135.1, 131.2, 126.5, 122.9, 120.0, 119.0, 116.4, 109.3, 106.6, 102.2, 100.4.

**MS (ESI^+^)**, m/z calcd for C_16_H_11_N_5_O_2_: 305.09, found 306.16 [M + H^+^].

#### 2.1.5. Synthesis of Lenalidomide Alkynylamide Linker-CRBN Ligand Construct **4**

Lenalidomide (500 mg, 1.93 mmol, 1.0 eq.) and 5-hexynoic acid (320 μL, 2.89 mmol, 1.5 eq.) were dissolved by stirring in dry DMF (16 mL) under N_2_ atmosphere at r.t.. HATU (734 mg, 1.93 mmol, 1.0 eq.) and DIPEA (1 mL, 5.79 mmol, 3 eq.) were sequentially added to the stirred solution. The resulting reaction mixture was stirred at r.t. for 24 h and monitored via TLC (eluent mixture: 9:1 DCM/MeOH, developed in phosphomolybdic acid ethanolic solution). After reaction completion, the mixture was quenched with H_2_O (20 mL). The aqueous phase was extracted with EtOAc (30 mL), and the collected organic phases dried over Na_2_SO_4_, and concentrated in vacuo. The crude pinkish solid was purified via flash chromatography (eluent mixture: from 92:8 to 95:5 DCM/MeOH), affording pure lenalidomide alkynylamide construct **4** (1.91 mmol, quantitative yield) as a white solid.

##### Analytical Characterization

**4**:



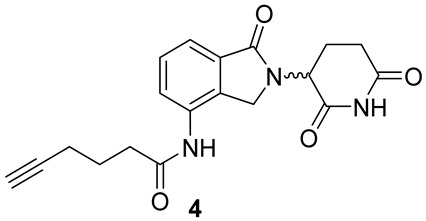



**^1^H NMR** (400 MHz, pyridine-d5): δ (ppm) = 12.90 (s, 1H), 10.82 (s, 1H), 7.97 (d, *J* = 7.9 Hz, 1H), 7.89 (d, *J* = 7.6 Hz, 1H), 7.46 (t, *J* = 7.7 Hz, 1H), 5.66 (dd, *J* = 13.4, 5.1 Hz, 1H), 4.83 (d, *J* = 16.9 Hz, 1H), 4.70 (d, *J* = 16.9 Hz, 1H), 2.99–2.86 (m, 1H), 2.88–2.79 (m, 1H), 2.77 (t, *J* = 7.3 Hz, 2H), 2.74 (t, *J* = 2.6 Hz, 1H), 2.43–2.28 (m, 3H), 2.13–2.04 (p, *J* = 7.1 Hz, 2H + m, 1H).

#### 2.1.6. Synthesis of Anle138b m-Triazole Connected Lenalidomide PROTAC **8a**

*m*/*p*-Azido Anle138b **2a** (50 mg, 0.16 mmol, 1.0 eq.) and alkyne **4** (70 mg, 0.20 mmol, 1.2 eq.) were dissolved in 1:1 DMF/H_2_O (8 mL) under stirring at r.t.. After sequential addition of CuSO_4_·5H_2_O (8 mg, 0.03 mmol, 0.2 eq.) and Na-ascorbate (30 mg, 0.16 mmol, 1.0 eq.), the solution was stirred at r.t. for 5 h. The reaction was monitored via TLC (eluent mixture: 9:1 DCM/MeOH, developed in phosphomolybdic acid ethanolic solution), and after its completion was quenched with H_2_O (10 mL). The aqueous phase was extracted with EtOAc (20 mL), and the collected organic phases were dried over Na_2_SO_4_, and concentrated in vacuo. The crude oil was purified via flash chromatography (eluent mixture: from 92:8 to 95:5 DCM/MeOH) or BIOTAGE^®^ reverse phase chromatography (eluent: H_2_O/ACN; from 0% ACN to 100% ACN), affording pure target PROTAC **8a** (0.09 mmol, 55% yield) as a bright yellow solid.

##### Analytical Characterization


**8a:**




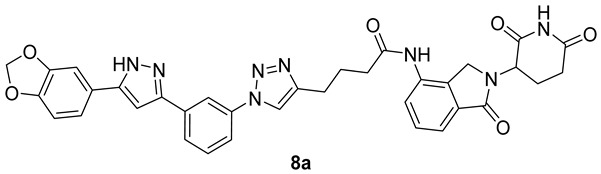



**^1^H NMR** (400 MHz, DMSO-d6): δ (ppm) = 13.27 (s, 1H), 11.01 (s, 1H), 9.91 (s, 1H), 8.73 (s, 1H), 8.36 (t, J = 1.9 Hz, 1H), 7.97–7.88 (m, 1H), 7.84 (m, 2H), 7.64 (t, J = 7.9 Hz, 1H), 7.55–7.44 (m, 2H), 7.42 (d, J = 1.7 Hz, 1H), 7.36 (dd, J = 8.1, 1.7 Hz, 1H), 7.28 (s, 1H), 7.02 (d, J = 8.0 Hz, 1H), 5.14 (dd, J = 13.3, 5.1 Hz, 1H), 4.64 (d, J = 6.5 Hz, 1H), 4.49–4.32 (m, 2H), 2.98–2.70 (m, 4H), 2.65–2.51 (m, 1H), 2.43–2.27 (m, 1H), 2.12–1.96 (m, 4H).

**^13^C NMR** (101 MHz, DMSO-d6): δ (ppm) = 172.9, 171.1, 167.9, 147.8, 147.6, 137.3, 133.8, 130.3, 128.6, 125.3, 124.7, 120.5, 119.0, 118.9, 116.2, 108.7, 105.6, 99.9, 51.6, 46.6, 31.2, 24.7, 24.6, 22.7.

**MS (ESI^+^)**, m/z calculated for C_35_H_30_N_8_O_6_: 658.23, found 659.54 [M + H^+^].

### 2.2. Biological Studies

#### 2.2.1. In Vitro Thioflavin T-Based Assay for αSyn Aggregation

Human αSyn (Sigma-Aldrich) was reconstituted with water to obtain a 5 mg/mL concentration. Tested samples were prepared, and their aggregation rate was checked with Thioflavin T as described in a published protocol [29]. Size exclusion chromatography buffer (20 mM K_2_HPO_4_, 5 mM KH_2_PO_4_, 100 mM KCl), then sodium azide (10% *w*/*v*), Thioflavin T (1 mM), and, lastly, αSyn (50 μM) were added into 1.5 mL reaction tubes. The samples were vortexed, then **Anle138b** and **8a**–**9b** were added in the reaction tubes, and the solutions were loaded in a 96-well plate. The Thioflavin T signal was measured for 6 days by taking three readings of fluorescence (excitation filter: 448-10; emission filter: 482-10) per day using a microplate reader (VICTOR3^TM^ PerkinElmer, Waltham, MA, USA). Quantification was performed by subtracting the mean value of each tested compound from the background fluorescence signal of Thioflavin T alone.

#### 2.2.2. Cytotoxic Effects of Anle138b-PROTAC Constructs on *4xSNCA* iPSC-Derived Neurons

DANs were generated as previously described [27]. iPSCs were dissociated with Accutase and plated on matrigel-coated 6-well plates (1 × 200.000 cells per well) in mTeSR1 medium. One day after, the medium was replaced by differentiation medium containing LDN193189 (100 nM, Stemgent), SB431542 (10 mM, Tocris), SHH C25II (100 ng ml21, R&D), purmorphamine (2 mM, Sigma-Aldrich), FGF8 (100 ng/mL, Sigma-Aldrich), and CHIR99021 (CHIR; 3 mM, Miltenyi Biotec, Bologna, Italy) in mTeSR1 medium for 11 days. The mTeSR1 medium was gradually shifted to N2 medium starting on day 5 of differentiation. Half medium was changed every 2–3 days. After 9 days, cells were dissociated with Accutase and plated on poly-L-lysine/laminin-coated 24-well plates for their final maturation. BDNF (10 ng/mL), GDNF (10 ng/mL), DAPT (10 μM, Sigma-Aldrich), and ascorbic acid (10 μM, Sigma-Aldrich) were added from day 20 to promote neuronal maturation and survival.

Neuronal cultures were seeded on matrigel-coated glass coverslips and treated for 7 days with Anle138b and **8a–9b**. Then, DANs were fixed for 20 min in ice in a 4% paraformaldehyde (PFA, Sigma) solution in PBS (Euroclone) and were permeabilized for 30 min in blocking solution, containing 0.5% Triton X-100 (Sigma-Aldrich) and 10% donkey serum (Sigma-Aldrich), and incubated overnight at 4 °C with primary antibodies in blocking solution. Then, cells were washed with PBS and incubated for 1 h at r.t. with Hoechst and with a secondary antibody. Anti-MAP2 (1:500, Immunological Sciences, Roma, Italy) was used as a primary antibody, while Alexa Fluor^TM^ was used as a secondary antibody for immunofluorescence staining. Images were collected using a X20/0.45 objective and analyzed using the ImageJ processing program.

#### 2.2.3. Cellular Assay for αSyn Aggregation on Patient-Derived DANs

Neurons from *4xSNCA* were seeded on matrigel-coated glass coverslips as previously reported [27]. Briefly, *4xSNCA* DANs were infected with AAV-FluoReSyn to visualize αSyn, and αSyn aggregation was assessed by GFP immunostaining. Cells were permeabilized for 30 min in blocking solution, containing 0.5% Triton X-100 (Sigma-Aldrich) and 10% donkey serum (Sigma-Aldrich), and incubated overnight at 4 °C with primary antibodies in blocking solution. Then, cells were washed with PBS and incubated for 1 h at r.t. with Hoechst and with a secondary antibody. Anti-GFP (1:500, ThermoFisher, Waltham, MA, USA) was used as primary antibody, while Alexa Fluor^TM^ was used as a secondary antibody for immunofluorescence staining. Cellular fluorescence images were acquired with an Olympus FV3000RS confocal microscope. Images were collected using a X63/1.4 oil objective and analyzed using Analyze Particles plugin in the ImageJ processing program.

## 3. Results

### 3.1. Medicinal Chemistry

#### 3.1.1. Rational Design and Synthesis of m/p Anle138b-Based Analogues **2a**,**b** Suitable for PROTAC Assembly

To build our PROTACs, we selected Anle138b as αSyn ligand due to its activity profile and drug-likeness, and to its known tolerance for aryl substitutions [14]. We also chose synthetically flexible thalidomide as a scaffold targeting CRBN E3 ligase [26]. We envisaged either a stable amide bond via amidation or a triazole bridge via ‘click’ chemistry as simple and biocompatible connecting chemistries. The former is widely used in biologically active compounds and the latter is an amide bioisoster shielded from in vivo protease hydrolysis [30]. Both reactions are high yielding, show excellent functional groups’ tolerability under mild reaction conditions, and allow the synthesis of small PROTACs arrays, provided the availability of suitable carboxylic acid/amine and azide/alkyne pairs [31]. Moreover, a biorthogonal ‘click’ chemistry could be exploited in cellulo/in vivo from smaller precursors, generating in situ the PROTAC hybrid [32] to prevent the putative scarce absorption for high molecular weight, pre-assembled PROTACs.

#### 3.1.2. Synthesis of m/p Anle138b Analogues **1a**,**b**, **2a**,**b** for PROTAC Assembly

We replaced the bromo substituent in Anle138b with an amine to provide direct access to an amide bond by coupling with suitable carboxylate linkers and indirect access to the bridge triazole ring via an azide formation—‘click’ chemistry sequence. We took advantage of a known procedure [33] for the efficient and rapid synthesis of 1,3-diketones from acyl chlorides with ketones. The resulting β-dicarbonyl intermediates were then converted in situ into the corresponding pyrazoles with hydrazine, as shown in Figure 1.

In detail, an excess of 3′- and 4′-nitroacetophenone enolate was generated with LiHMDS at 0 °C (step a1), then the acyl chloride was added to obtain a Claisen condensation unstable β-diketones (step a2), which were converted in situ into pyrazoles **3a**,**b** upon reaction with hydrazine and acetic acid (step b). The nitro group was then reduced in standard Fe powder–H_2_SO_4_ conditions, yielding target Anle138b-based anilino derivatives **1a**,**b** in overall good yields (step c). Finally, a portion of anilino derivatives **1a**,**b** was converted into the corresponding azides **2a**,**b** in very good yields by formation of the diazonium salt with sodium nitrite in acid conditions and in situ displacement with sodium azide (step d, Figure 1).

#### 3.1.3. Synthesis of Thalidomide-/Lenalidomide-Linker Constructs **4**–**6** for Coupling with m/p Anle138b Analogues

Commercial lenalidomide and fluoro-thalidomide were coupled respectively via amide condensation or aryl nucleophilic displacement with suitable commercial alkyne- or carboxylate-bearing linkers, as shown in Figure 2. In detail, alkynylamide **4** was quantitatively obtained by standard amide condensation between 5-hexynoic acid and lenalidomide (step a, Figure 2). Conversely, a slight excess of PEGylated alkynylamine and 6-aminohexanoic acid was used in an aryl basic nucleophilic displacement with fluoro-thalidomide to yield alkynylamine **5** and carboxyalkyl amine **6** (respectively steps b and c, Figure 2) in poor, unoptimized yields.

It is worth mentioning that linker-CRBN ligand constructs **4**, **5**, and **6** may be used in future efforts in our laboratories, coupling them to known ligands of aggregation-prone, ND/cancer-connected proteins other than αSyn.

#### 3.1.4. PROTAC Hybrid Assembly: Click Reaction- and Amide-Based Strategies to **7a**,**b**–**9a**,**b**

Target amide-connected Anle138b-PROTACs **7a**,**b** (amide-lenalidomide) were synthesized in moderate yields via standard amide condensation between anilino derivatives **1a**,**b** and linker-CRBN carboxylate construct **6** (step a, Figure 3).

Target triazole-connected Anle138b-PROTACs **8a**,**b** (triazole-amide-lenalidomide) and **9a**,**b** (triazole-PEGylated amine-thalidomide) were assembled by reacting azido derivatives **2a**,**b** with linker-CRBN alkynyl constructs **4** and **5**, respectively, under classical ‘click’ reaction conditions as reported in step a, Figure 4. Poor to excellent unoptimized reaction yields were observed.

Four triazole-connected putative Anle138b-PROTAC hybrids **8a**,**b**, **9a**,**b** were submitted to preliminary biological studies, as shown in the next Section.

### 3.2. Biology Studies

#### 3.2.1. In Vitro αSyn Aggregation Assay

A Thioflavin T (ThT) cell-free fluorescence assay [28] was carried out in duplicates to determine the binding of standard Anle138b, alkyl-connected *m*- and *p*-triazole amide PROTACs **8a**,**b**, and PEG-connected *m*- and *p*-triazole amide PROTACs **9a**,**b** (500 nM) to αSyn monomers/oligomers (50 μM) to inhibit their aggregation. αSyn aggregation was measured at ten timepoints in a six days timespan. As a control, an assay without any putative inhibitor was also run in duplicates. Results are shown in Figure 1.

The four putative PROTAC compounds bind to αSyn monomers/oligomers and inhibit their further aggregation, although to a lesser extent than parent Anle138b. Their measured effect on aSyn aggregation is reliable, as the SD is small at later time points/highest compound effect. Among them, lenalidomide-containing *p*-PEGylated amine **9b** (red) and thalidomide-containing *m*-amide **8a** (orange curve) display the greatest reduction in fluorescence emission.

#### 3.2.2. Cellular Assay for αSyn Aggregation on Patient-Derived Dopaminergic Neurons

Patient-derived dopaminergic neuronal cultures bearing four αSyn *SNCA* gene copies (*4xSNCA* DANs) were used to determine the effect of Anle138b-based PROTAC constructs **8a**,**b** and **9a**,**b** on αSyn aggregation with a nanobody biosensor, following a published procedure [27]. Prior to that, the constructs were checked for their effects on neurons (Figure 2). Neuronal cultures were treated for 7 days with Anle138b and putative PROTACs **8a**–**9b**, exchanging the medium every other day, and MAP2-positive neurons were then counted. Surprisingly, only thalidomide-containing *m*-amide **8a** was as safe as Anle138b and could be further evaluated, while compounds **8b**, **9a** and **9b** promoted detachment and loss of most cultured DANs (Figure 2).

Then, *4xSNCA* DANs were incubated for a week with **8a** or Anle138b (10 µM), treated with the nanobody-based fluorescent reporter for hαSyn biosensor (FluoReSyn) [27], and incubated for another week (Figure 3, both as images and histograms). Neurons exposed to **8a** for two weeks (bottom lane) showed a robust reduction in fluorescence and aggregates per soma (orange histogram, 11.6 ± 0.3 aggregates, 52 ± 3%, Figure 3) with respect to untreated cells (top lane and blue histogram, 20.8 ± 0.3 aggregates, 100%). Since FluoReSyn is a reliable sensor of intracellular αSyn levels, this clearly indicates a significant reduction of total αSyn content in cells treated with **8a**, as FluoReSyn equally binds to and visualizes both monomeric as well as oligomeric αSyn forms [27]. Remarkably, Anle138b showed a lower effect than **8a** on *4xSNCA* DANs (middle lane and red histogram, 15.9 ± 0.4 aggregates, 76 ± 4%, Figure 3).

## 4. Discussion and Conclusions

Dual-action compounds designed to address two pathological mechanisms in complex, multifactorial diseases may represent a promising therapeutic avenue [34]. Once two validated mechanisms are selected, a bifunctional putative modulator must be designed and synthesized by connecting two active fragments through a linker. Preservation of target affinity for both fragments and bioavailability (i.e., limited size and acceptable partition coefficient/LogP) must be ensured also through flexible linkers with variable length, rigidity, and hydrophilicity [35].

We targeted αSyn aggregation as a validated intervention pathway through small molecule inhibition, to therapeutically deal with synucleinopathies [10]. An assessed mechanism, represented by αSyn binders preventing aggregation, was exploited through clinically tested Anle138b [14,15]. αSyn binding was exploited also through the validated, target protein-degrading PROTAC mechanism and CRBN-binding lenalidomide and thalidomide [26].

Six αSyn-targeted putative PROTACs **7a**,**b**–**9a**,**b** were designed and synthesized. Their preliminary evaluation showed structure-dependent binding to αSyn in a cell-free model. Major differences in biological activities were observed in cellular models in terms of undesirable cytotoxicity, and on an aggressive early parkinsonism’ model [27]; they surely stem from Anle138b substitution patterns (compare cytotoxic *p*-substituted **8b** with αSyn-reducing *m*-substituted **8a**), and on linkers’ nature and length (compare cytotoxic triazole-PEGamine-connected **9a** with αSyn-reducing triazole-amide **8a**). Further structural diversification of αSyn-targeted, Anle138b-centered PROTACs is needed to better rationalize the observed preliminary structure-activity relationship (SAR), although m-substitution patterns (as in **8a**) will be important in future efforts.

The significant αSyn-reducing effect exerted by Anle138b-based PROTAC construct **8a** is also worth further study. The marked decrease in αSyn levels caused by **8a** in *4xSNCA* DANs could be attributed to a synergistic action between the known αSyn binding/anti-aggregation effect (Anle138b being more potent than **8a**, Figure 1) with the putative and **8a**-specific UPS-promoted degradation of αSyn. Such dual action likely provides a stronger impact on misfolded/aggregated αSyn-dependent pathologies. Thus, we plan to report in future data about CRBN binding-of **8a**, promoting αSyn-degradation, and about its eADMET properties. We also plan to test **8a** (or an improved analogue to follow) in an in vivo model of neurodegenerative synucleinopathies, but also in an oncology model, where αSyn, or structurally related γSyn [36], plays a major role in disease progression.

## Data Availability

Not applicable.

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
