# Peer review of "Synthesis and Preliminary Characterization of Putative Anle138b-Centered PROTACs against α-Synuclein Aggregation"

_pharmaceutics, 2023, doi:10.3390/pharmaceutics15051467_

Round 1
Reviewer 1 Report
Authors of this manuscript reported the development of Anle138b-centered PROTACs against a-synuclein aggregation. A small array of six PROTACs were synthesized based on a previously reported aSyn ligand Anle138b. Cytotoxic effects of the compounds were determined. aSyn aggregation was determined by in vitro Thioflavin T-based assay and cellular assay. Overall, the study will be of interest to the medicinal chemistry community. However, several issues need to be addressed.
Major issues are listed in the following.
1. There are reported peptide PROTACs as well as PROTAC compounds targeting a-synuclein invented by Arvinas. These compounds should be mentioned in the introduction section.
2. The drug design part should be more rational.
3. Biological data was very limited. Degradation of a-synuclein should be evaluated.
4. There are only six compounds in the array. The structure-affinity relationship should be discussed in detail. Why minor difference in chemical structures leads to big difference in a-synuclein aggregation?
5. All 13C NMR spectra in the SI should be corrected. Phase corrections should be done.
The manuscript is well written and clearly presented.
Author Response
The revised manuscript takes into account all the pertinent observations and suggestions raised by four Referees; such observations are also addressed in a single file for all, attached here, with highlighted answers in red. Only a single, requested 13C NMR spectrum (compound 8b, point 5, Referee 3) is to be added to the Supplementary Information file, and will be recorded by May 5th; it will be replaced in the SuppInfo file, and such final version will separately be sent to the Editor, to avoid delays.

Reviewer 2 Report
The present manuscript entitled "Synthesis and preliminary characterization of putative Anle138b-Centerd PROTACs against α-Synuclein Aggregation" by Pedrini et al involved the search for novel agents against Parkinson disease, specifically regarding the PROTAC hybrids biological targets.
The introduction is complete and according to the developed topic of the manuscript, and it has updated bibliographical references to support the research.
Also, the manuscript is interesting, clear, organize, and focused on the topic that is of interest in the medicinal chemistry field of research.
Furthermore, the results are organized, and they are analyzed and discussed according to the relevant obtained results.
Moreover, the information they described is supported with clear and logical figure that summarize all the required data for the discussion item.
Furthermore, some typo mistakes should be corrected (in yellow, pdf file attached) such as:
· It should be necessary the authors clarify why they choose these molecules to synthesized due to the lack of explanation regarding the molecules core or the substitution used for them. The lines 248-251 are not enough to understand the rational design behind the selected molecules.
· Line 322. The authors declare the assays were done by duplicate. Please explain why they didn’t use triplicates in order to obtain the SD parameters for these analyses (figures and data without SD)
· Considering the same analysis, why the authors didn’t use a positive control for the assay (to understand the impact of the reported results).
· Please remember that the unit and the number (e.g. 20 nM – 30 °C – 10 min – 2 h – 2 Å) should have a blank space between them. Please check them all throughout the manuscript.
· Please remember to use italics for in vitro, in vivo and in silico.
Finally, I would like to invite the authors to include the abbreviation list of words at the end of this manuscript.

Author Response

(The authors gave the same response as above.)

Reviewer 3 Report
The authors describe the synthesis of six Anle138b-centered PROTACs targeting α-Synuclein. Although the manuscript is interesting, it needs major revision.
Abstract lines 18, 22 and 27: the word hybrids is redundant
Introduction lines 80-85: this part of introduction should describe aim of the studies, not their results.
Experimental part
For several compounds, the description of the number of protons or carbons does not correspond to the molecular formula. The authors should recheck the entire spectral characterization of reported compounds.
Molybdic reagent: it is not a correct name.
For compounds 7a and 8b the 13C MNR spectra should be taken at higher concentration.
Lines 219 and 315: replace “profiling” by a more appropriate word ie studies.
Lines 272, 285: replace “in details” by “in detail”
Conclusion line 381: “its marked decrease in αSyn levels caused by 8a could be attributed to an autophagy-dependent degradation of αSyn” The authors should clarify this point. Autophagy-targeting chimeras (AUTACs) represent a different targeted protein degradation (TPD) strategy, not involving the ubiquitin–proteasome system.
Moderate editing of English Language is needed
Author Response

(The authors gave the same response as above.)

Reviewer 4 Report
This study examines the potential use of six Anle138b-centered PROTAC hybrids targeting a-Synuclein aggregates for binding, polyubiquitination by the E3 ligase Cereblon, and proteasomal degradation. Four Anle138b-PROTAC hybrids (8 a,b and 9 a,b) were characterized against in vitro αSyn aggregation, monitoring them in a Thioflavin T fluorescence assay and in dopaminergic neurons derived from a set of isogenic pluripotent stem cell lines with SNCA multiplications.
As a disclaimer, the chemical part of this research is far out of my domain of expertise. Therefore, no comments will be forwarded on that part. Overall the authors provided a really interesting and comprehensible paper. However, some plot holes might be present and will be described in the different comments below:
- Anle138b has already been shown to bind to pathological aggregates and strongly inhibited formation of pathological oligomers α-synuclein in vitro and in vivo (literature + Fig 1.). Moreover, Anle138b already passed the phase 1a of clinical trial showing good tolerance in healthy humans. However, Anle138b has never been shown to increase the process of α-synuclein aggregates. Therefore, I would suggest to the authors to emphasize on the novelty of their research which implies the creation of an Anle138b hybrid which is able to reduce α-synuclein aggregation (although to a lesser extend than the original one (Fig 1.) but also to increase a-synuclein proper processing.
- Fig 2: Even though the images are pretty clear, an independent live/dead assay with quantification could be used to complement the neurotoxicity induced by the different PROTAC constructs and strengthen the results.
- Fig 3: Suddenly, the control Anle138b disappear from the analysis. We can clearly see the improvement compared to the basal condition but we can’t say whether it’s due to the PROTAC complex or just because of the presence of Anle138b. The authors should add the same analysis with the Anle138b threated condition to strengthen their results as they did for the figure 2.
Author Response

(The authors gave the same response as above.)

Round 2
Reviewer 1 Report
The authors have addressed all concerns by this reviewer.
Reviewer 3 Report
The manuscript is suitable for publication in Pharmaceutics